# A multiple risk factor program is associated with decreased risk of cardiovascular disease in 70-year-olds: A cohort study from Sweden

**Anna Nordström** [1,2]*, **Jonathan Bergman** [3], **Sabine Björk** [1], **Bo Carlberg** [4], **Jonas Johansson** [5], **Andreas Hult** [1], **Peter Nordström** [3]

**1** Division of Sustainable Health, Department of Public Health and Clinical Medicine, Umeå University, Umeå, Sweden, **2** School of Sport Sciences, The Arctic University of Norway, Tromsø, Norway, **3** Unit of Geriatric Medicine, Department of Community Medicine and Rehabilitation, Umeå University, Umeå, Sweden, **4** Division of Medicine, Department of Public Health and Clinical Medicine, Umeå University, Umeå, Sweden, **5** Department of Community Medicine, The Arctic University of Norway, Tromsø, Norway

* anna.h.nordstrom@umu.se

## Abstract

### Background

In individuals below 65 years of age, primary prevention programs have not been successful in reducing the risk of cardiovascular disease (CVD) and death. However, no large study to our knowledge has previously evaluated the effects of prevention programs in individuals aged 65 years or older. The present cohort study evaluated the risk of CVD in a primary prevention program for community-dwelling 70-year-olds.

### Method and findings

In 2012–2017, we included 3,613 community-dwelling 70-year-olds living in Umeå, in the north of Sweden, in a health survey and multidimensional prevention program (the Healthy Ageing Initiative [HAI]). Classic risk factors for CVD were evaluated, such as blood pressure, lipid levels, obesity, and physical inactivity. In the current analysis, each HAI participant was propensity-score-matched to 4 controls ($n$ = 14,452) from the general Swedish population using national databases. The matching variables included age, sex, diagnoses, medication use, and socioeconomic factors. The primary outcome was the composite of myocardial infarction, angina pectoris, and stroke. The 18,065 participants and controls were followed for a mean of 2.5 (range 0–6) years. The primary outcome occurred in 128 (3.5%) HAI participants and 636 (4.4%) controls (hazard ratio [HR] 0.80, 95% CI 0.66–0.97, $p$ = 0.026). In HAI participants, high baseline levels of blood pressure and lipids were associated with subsequent initiation of antihypertensive and lipid-lowering therapy, respectively, as well as with decreases in blood pressure and lipids during follow-up. In an intention-to-treat approach, the risk of the primary outcome was lower when comparing all 70-year-olds in Umeå, regardless of participation in HAI, to 70-year-olds in the rest of Sweden for the first 6 years of the HAI project (HR 0.87, 95% CI 0.77–0.97, $p$ = 0.014). In contrast, the risk was similar in the 6-year period before the project started (HR 1.04, 95% CI 0.93–1.17, $p$ = 0.03 for

**Data Availability Statement:** Researchers, including international researchers, who are

interested in obtaining the data can contact SCB via information@scb.se, and Socialstyrelsen via registerservice@socialstyrelsen.se. One can visit https://www.scb.se/sv_/Vara-tjanster/Bestalla-mikrodata/ (SCB), http://www.socialstyrelsen.se/register/bestalladatastatistik/bestallaindividuppgifterforforskningsandamal (Socialstyrelsen).

**Funding:** AN and PN received grants from the Swedish research council for the present study. The funder had no role in study design, data collection and analysis, decision to publish, or preparation of the manuscript.

**Competing interests:** The authors have declared that no competing interests exist.

**Abbreviations:** CVD, cardiovascular disease; HAI, Healthy Ageing Initiative; HR, hazard ratio; LDL, low-density lipoprotein; NCD, noncommunicable disease; WHO, World Health Organization.

interaction). Limitations of the study include the observational design and that changes in blood pressure and lipid levels likely were influenced by regression towards the mean.

## Conclusions

In this study, a primary prevention program was associated with a lower risk of CVD in community-dwelling 70-year-olds. With the limitation of this being an observational study, the associations may partly be explained by improved control of classic risk factors for CVD with the program.

## Author summary

### Why was this study done?

- In individuals below 65 years of age, primary prevention programs have not been successful in reducing the risk of cardiovascular disease (CVD) and death. However, no previous large study to our knowledge has evaluated the effects of a primary prevention program in individuals aged 65 years or older.

### What did the researchers do and find?

- In total, 3,613 70-year-olds were included in a health survey and multidimensional prevention program. Each individual was propensity-score-matched to 4 controls from the Swedish population. Finally, individuals were followed-up for CVD.

- During a mean follow-up time of 2.5 years, the prevention program was associated with 20% lower risk of CVD (hazard ratio [HR] 0.80, 95% CI 0.66–0.97, $p = 0.026$).

- During the years of the prevention program, the risk of CVD was also 13% lower (HR 0.87, 95% CI 0.77–0.97, $p = 0.014$) for 70-year-olds overall in the municipality where the prevention program took part, compared to in the rest of Sweden. In contrast, the risk of CVD was 4% higher in the municipality than the rest of Sweden before the prevention program started.

### What do these findings mean?

- With the limitation of the observational design, the findings suggest that prevention programs could reduce the risk of CVD in older individuals by targeting classic risk factors, such as blood pressure.

## Introduction

The leading causes of death and morbidity worldwide are noncommunicable diseases (NCDs), such as chronic respiratory disease, cancer, diabetes, and, in particular, ischemic heart disease and stroke [1]. The impact of these NCDs will probably increase with a growing number of

older people in the future [2], imposing great challenges on care systems. However, many NCDs share modifiable risk factors such as smoking, substance abuse, unhealthy diet, and physical inactivity. The World Health Organization (WHO) has concluded that stroke, heart disease, and type 2 diabetes can be prevented by lifestyle change in at least 80% of the individuals affected [3]. In addition, appropriate pharmacological interventions decrease the impact of risk factors such as hypertension and hyperlipidemia.

Thus, there are several incentives to evaluate strategies that may reduce the risk of our most common NCDs. Although preventive measures likely are necessary cornerstones of these strategies, a meta-analysis of randomized controlled trials showed that primary prevention programs involving education and counseling did not reduce the risk of cardiovascular disease (CVD) and death [4]. This lack of effect may relate to the facts that the mean age in the included studies was only 50 years and that the risk factor burden was rather low. The lack of effect in previous studies may also be related to short follow-up times, since preventive measures in midlife may influence the risk of disease many years later. To the best of our knowledge, no large study has previously evaluated the effects of prevention programs in individuals aged 65 years or older, in whom risk factors for CVD are much more common. The aim of the present study was to evaluate the effects of a multiple risk factor program—including collection and evaluation of classic risk factors for CVD such as blood pressure, lipid levels, obesity, and physical inactivity, and feedback to participants—on the risk of ischemic heart disease and stroke in 70-year-old men and women.

## Methods

### Healthy Ageing Initiative program

The Healthy Ageing Initiative (HAI) is an ongoing primary prevention study in Umeå, a municipality with 127,000 inhabitants in northern Sweden. The study was initiated in May 2012, and is performed at a single clinic. Three trained research nurses conduct all the testing, with the support of 2 chief physicians (AN and PN). The eligibility criteria are residence in Umeå Municipality and age exactly 70 years. There are no exclusion criteria, and public population registers are used for recruitment. During the years of this study, 54% of 70-year olds in Umeå have participated.

The prevention program used in the HAI project is described in detail in S1 Appendix. In short, all participants arrive fasting at a first visit for measurements of blood glucose and lipids. Other data collected include a comprehensive self-administered health and lifestyle questionnaire; total, gynoid, and visceral fat mass, measured using dual-energy X-ray absorptiometry; waist and hip circumference, measured using a measuring tape; and blood pressure, which is measured in a seated position after at least 10 minutes of rest. Participants are given feedback on their test results (e.g., blood pressure, BMI, and blood glucose) based on cut points from current guidelines. In total this first visit takes about 3 hours. The participants are then sent home with an accelerometer for assessment of physical activity during 1 week. Thereafter, the participants return for a second visit at which all test results form the basis for a motivational interview about diet, exercise, and tobacco and alcohol use. In addition, participants are encouraged to contact their general practitioner for appropriate medication adjustments.

### Participants and controls

The aim of the present cohort analysis was to evaluate the HAI prevention program with respect to CVD. The study protocol was registered on ClinicalTrials.gov on 17 October 2017 (NCT03312439), and we received data files for the present project from the National Board of Health and Welfare on 21 December 2018. We hypothesized that the prevention program

would reduce the primary outcome of CVD. However, some analyses, such as those on blood pressure and lipid levels, were added in response to reviewers' comments. To evaluate the prevention program, in the primary analysis, we included everyone who participated in HAI during 2012–2017, in total 3,617 individuals, of whom 3,613 had complete data. We created a control group from the general population of Sweden using national registry data. Using the Register of the Total Population [5], we identified controls who resided in Sweden on 31 December 2005 and who were expected to turn 70 years during the analysis period.

In HAI participants, baseline was the date of the HAI health survey. To assign a baseline date to controls, we randomly sampled the time interval between the HAI participants' 70th birthday and the date of their visit. Next, these sampled intervals were added to the controls' 70th birthday. Controls were excluded from the analysis if they were not alive on their assigned baseline date, if they had emigrated, or if this date was not in 2012–2017.

In a secondary analysis, the risk of the outcome was compared between all 70-year-old Umeå residents, irrespective of participation in the HAI project, and 70-year-olds residing in the rest of Sweden who turned 70 years during 2006–2017. This analysis was divided into 2 periods: the 6 years before the start of HAI (2006–2011) and the first 6 years of the study (2012–2017). In both periods, baseline dates for non-HAI participants were randomly assigned as described above.

## Confounders

Data on diagnosed medical and psychiatric conditions were collected from the National Patient Register, a register managed by the National Board of Health and Welfare that covers all inpatient care in Sweden since 1987 and all specialist outpatient care since 2001 [6]. Data on prescription medication use were obtained from the Prescribed Drug Register, which covers all medications sold in Sweden since July 2005. Socioeconomic data (income, education, and civil status) were collected from the registers of Statistics Sweden and the National Board of Health and Welfare [5]. Detailed variable definitions are provided in S1 Table.

## Outcomes

The primary outcome was the occurrence of ischemic or hemorrhagic stroke, myocardial infarction, or angina pectoris until 31 December 2017. These events were traced through the National Patient Register using diagnostic codes I20, I21, and I61–I64 (International Classification of Diseases–10th Revision). In addition, for HAI participants, changes in blood pressure and lipid levels and prescription of antihypertensive drugs and lipid-lowering therapy were investigated after the baseline examination. Blood pressure measurements and low-density lipoprotein (LDL)–cholesterol were obtained by using a script to scan all medical records in primary and specialized healthcare, based on the unique personal identity number given to all Swedish residents.

## Statistical analysis

In the primary analysis, we matched each HAI participant to 4 controls, matching exactly on sex and year of birth, and propensity-score-matching on diagnoses, socioeconomic variables, and prescription medication use (all variables in Table 1). Propensity scores were estimated using logistic regression. This model included a square term for disposable income, the only continuous covariate. The propensity-score match was created using a nearest-neighbor algorithm without replacement (Psmatch2 package for Stata). Match quality (closeness) was determined using standardized mean differences, where differences of <0.1 were considered negligible [7]. We compared the risk of the primary outcome in HAI participants and matched controls using Cox regression, a model that was stratified by matched set.

**Table 1. Baseline characteristics of Healthy Ageing Initiative (HAI) participants and the control population before and after matching.**

| Variable | HAI participants | Controls | | Standardized mean difference | |
|---|---|---|---|---|---|
| | | Before matching | After matching | Before matching | After matching |
| *N* | 3,617 | 653,747 | 14,452 | | |
| **Age, mean (SD), years** | 70.5 (0.1) | 70.5 (0.1) | 70.5 (0.1) | 0.01 | 0.01 |
| **Female sex, *n* (%)** | 1,824 (50.5) | 329,373 (50.7) | 9,120 (50.5) | 0.01 | 0.00 |
| **Disposable income at age 60 years, mean (SD), 1,000 Swedish kronor** | 245 (175) | 235 (580) | 249 (226) | 0.02 | 0.02 |
| Missing data, *n* | 0 | 462 | | | |
| **Education[a], *n* (%)** | | | | | |
| Primary | 613 (16.9) | 194,956 (29.8) | 2,387 (16.5) | 0.34 | 0.02 |
| Secondary | 1,481 (40.9) | 278,087 (42.5) | 6,104 (42.2) | 0.03 | 0.03 |
| Post-secondary | 1,519 (42.0) | 176,329 (27.0) | 5,961 (41.1) | 0.27 | 0.02 |
| Missing data, *n* | 4 | 4,375 | | | |
| **Civil status[a], *n* (%)** | | | | | |
| Married | 2,383 (65.9) | 395,790 (60.5) | 9,529 (65.9) | 0.11 | 0.00 |
| Never married | 317 (8.8) | 71,967 (11.0) | 1,233 (8.5) | 0.08 | 0.01 |
| Widowed | 284 (7.9) | 55,644 (8.5) | 1,106 (7.7) | 0.02 | 0.01 |
| Divorced | 633 (17.5) | 129,313 (19.8) | 2,584 (17.9) | 0.06 | 0.01 |
| Other, *n* | 0 | 266 | | | |
| Missing data, *n* | 0 | 767 | | | |
| **Diagnoses, *n* (%)** | | | | | |
| Stroke | 122 (3.4) | 27,215 (4.2) | 481 (3.3) | 0.04 | 0.00 |
| Myocardial infarction | 168 (4.6) | 32,515 (5.0) | 664 (4.6) | 0.02 | 0.00 |
| Heart failure | 63 (1.7) | 16,115 (2.5) | 240 (1.7) | 0.06 | 0.01 |
| Angina pectoris | 282 (7.8) | 47,403 (7.3) | 1,125 (7.8) | 0.02 | 0.00 |
| Diabetes | 326 (9.0) | 88,836 (13.6) | 1,368 (9.5) | 0.16 | 0.02 |
| Fracture | 570 (15.8) | 96,327 (14.7) | 2,314 (16.0) | 0.03 | 0.01 |
| Rheumatoid arthritis | 79 (2.2) | 12,629 (1.9) | 326 (2.3) | 0.02 | 0.00 |
| Chronic obstructive pulmonary disease | 61 (1.7) | 18,912 (2.9) | 244 (1.7) | 0.09 | 0 |
| Renal failure | 27 (0.7) | 9,411 (1.4) | 125 (0.9) | 0.08 | 0.02 |
| Crohn disease | 29 (0.8) | 5,200 (0.8) | 107 (0.7) | 0.00 | 0.01 |
| Ulcerative colitis | 31 (0.9) | 5,803 (0.9) | 139 (1.0) | 0.00 | 0.01 |
| Parkinson disease | 26 (0.7) | 4,127 (0.6) | 91 (0.6) | 0.01 | 0.01 |
| Dementia | 14 (0.4) | 6,865 (1.1) | 47 (0.3) | 0.11 | 0.01 |
| Depression | 735 (20.3) | 144,008 (22.0) | 2,947 (20.4) | 0.04 | 0.00 |
| Bipolar disorder | 18 (0.5) | 4,102 (0.6) | 66 (0.5) | 0.04 | 0.01 |
| Alcohol intoxication | 31 (0.9) | 15,335 (2.3) | 97 (0.7) | 0.16 | 0.02 |
| Opioid intoxication | 1 (0.03) | 692 (0.1) | 2 (0.01) | 0.05 | 0.01 |
| Cancer | 676 (18.7) | 125,498 (19.2) | 2,722 (18.8) | 0.01 | 0.00 |
| **Medications[b], *n* (%)** | | | | | |
| Antihypertensive | 2,083 (57.6) | 357,358 (54.7) | 8,490 (58.7) | 0.06 | 0.02 |
| Lipid-lowering agent | 1,558 (43.1) | 256,229 (39.2) | 6,260 (43.3) | 0.08 | 0.00 |
| Anticoagulant | 1,412 (39.0) | 257,188 (39.3) | 5,641 (39.0) | 0.01 | 0.00 |
| Neuroleptic | 82 (2.3) | 20,793 (3.2) | 300 (2.1) | 0.06 | 0.01 |
| Hypnotic | 955 (26.4) | 176,796 (27.0) | 3,861 (26.7) | 0.01 | 0.01 |
| Sedative | 378 (10.5) | 133,857 (20.5) | 1,535 (10.6) | 0.33 | 0.01 |
| Immunosuppressant | 110 (3.0) | 20,357 (3.1) | 444 (3.1) | 0.00 | 0.00 |

[a]Education and civil status recorded in the calendar year before the baseline date.

[b]Prescriptions filled since July 2005.

In the secondary analysis of 70-year-olds in Umeå and those in the rest of the Sweden, associations were investigated using unconditional Cox regression. To test whether the associations were different in the periods before and after HAI started, an interaction term was created between time period of baseline date (2006–2011 or 2012–2107) and whether individuals were Umeå residents (yes or no). The proportional hazards assumption was evaluated for the models by scaled Schoenfeld residuals. Statistical analyses were performed using Stata version 15.0 (StataCorp, College Station, TX, US) and SPSS version 25.0 (IBM, Armonk, NY, US). $p$-Values $< 0.05$ were considered to be significant.

## Data linkage and ethics approval

HAI data and national registry data could be linked by unique personal identity numbers, issued to all residents of Sweden upon birth or immigration. The HAI study and the present analysis were both approved by the Regional Ethical Review Board in Umeå, Sweden (no. 07-031M with extensions). Written informed consent was given by all participants. This paper follows the STROBE reporting guideline (S1 STROBE Checklist).

## Results

### Study cohort

For the primary analysis, data were available for 3,617 HAI participants (of whom 3,613 had complete data) and 734,359 potential controls born in 1942–1947. Potential controls were excluded ($n = 80,612$) if they emigrated before the assigned baseline date, if the baseline date was out of range, i.e., not in 2012–2017, or if the baseline date was after their date of death. Thus, there were 3,613 eligible HAI participants and 653,747 eligible controls. Propensity-score matching resulted in a final cohort of 3,613 HAI participants and 14,452 controls with similar baseline characteristics (Table 1). Additional baseline characteristics for participants, collected in the HAI health study are provided in Table 2. Baseline characteristics for all Umeå residents are presented in Table 3. Individuals who participated in HAI were generally healthier, with lower prevalence of CVD and diabetes, than Umeå residents who did not participate.

### Outcome of stroke, myocardial infarction, or angina pectoris

The matched cohort ($n = 18,065$) was followed for a mean of 2.5 years (range 0–6 years) (Fig 1). During follow-up, the primary outcome of stroke, myocardial infarction, or angina pectoris occurred in 128 (3.5%) HAI participants and 636 (4.4%) controls (hazard ratio [HR] 0.80, 95% CI 0.66–0.97, $p = 0.026$). The HR was similar in the male (HR 0.78, 95% CI 0.62–0.99, $p = 0.046$) and female (HR 0.79, 95% CI 0.57–1.11, $p = 0.31$) subcohorts.

In a secondary analysis, we compared the risk of the primary outcome in all 70-year-old Umeå residents to that in 70-year-old individuals in the rest of Sweden (Table 3). At baseline, the groups were similar in most respects, the exceptions being education and use of antihypertensives, lipid-lowering agents, and sedatives. In the 6-year period before HAI started (years 2006–2011), the primary outcome occurred in 284 (6.5%) Umeå residents, compared to 27,274 (6.1%) individuals from the rest of Sweden (HR 1.06, 95% CI 0.94–1.19, $p = 0.33$; Fig 2). This association changed marginally after adjusting for all confounders (HR 1.04, 95% CI 0.93–1.17, $p = 0.51$). In contrast, during the first 6 years of the HAI prevention program (years 2012–2017), in which 54% of Umeå residents participated, the outcome occurred in 291 (4.4%) Umeå residents and 31,851 (4.9%) residents of the rest of Sweden (HR 0.87, 95% CI 0.77–0.97, $p = 0.03$ for interaction; Fig 3), after adjustment for all confounders.

**Table 2. Additional baseline characteristics collected from participants in the Healthy Ageing Initiative study.**

| Variable | Total cohort (*n* = 3,617) | Women (*n* = 1,817) | Men (*n* = 1,800) |
|---|---|---|---|
| **Body composition** | | | |
| Height (cm) | 170 ± 9 | 163 ± 6 | 176 ± 6 |
| Weight (kg) | 77.0 ± 15.0 | 70.3 ± 13.0 | 83.7 ± 12.9 |
| BMI (kg/m$^2$) | 26.6 ± 4.3 | 26.4 ± 4.7 | 26.8 ± 3.8 |
| Waist circumference (cm) | 94 ± 13 | 89 ± 12 | 99 ± 11 |
| Hip circumference (cm) | 103 ± 8 | 103 ± 9 | 102 ± 7 |
| Total fat mass (grams)* | 27,645 ± 9,089 | 28,887 ± 9,364 | 26,389 ± 8,624 |
| Gynoid fat mass (grams)* | 4,020 ± 1,418 | 4,604 ± 1,434 | 3,429 ± 1,128 |
| Android fat mass (grams)* | 2,748 ± 1,190 | 2,586 ± 1,158 | 2,912 ± 1,199 |
| **Blood pressure** | | | |
| Systolic (mm Hg) | 139 ± 17 | 140 ± 17 | 138 ± 16 |
| Diastolic (mm Hg) | 81 ± 9 | 81 ± 9 | 82 ± 9 |
| **Blood glucose (mmol/l)** | 5.7 ± 1.2 | 5.6 ± 1.3 | 5.8 ± 1.2 |
| **Blood lipids (mmol/l)** | | | |
| Total cholesterol | 5.4 ± 1.2 | 5.8 ± 1.1 | 5.1 ± 1.2 |
| Low-density lipoprotein cholesterol | 3.3 ± 1.1 | 3.5 ± 1.1 | 3.0 ± 1.0 |
| High-density lipoprotein cholesterol | 1.6 ± 0.5 | 1.7 ± 0.5 | 1.4 ± 0.4 |
| Triglycerides | 1.3 ± 0.7 | 1.3 ± 0.7 | 1.4 ± 0.7 |
| **Accelerometer-measured physical activity (steps/day)** | 7,331 ± 3,093 | 7,298 ± 3,132 | 7,364 ± 3,052 |
| **Current smoker (*n*, %)** | 214, 5.9% | 119, 6.5% | 95, 5.3% |

Except where otherwise noted, data are mean ± standard deviation.

*Measured by dual-energy X-ray absorptiometry.

## Blood pressure and LDL-cholesterol in HAI participants

In the 3,617 HAI participants, 53.6% had hypertension stage 2 at baseline (blood pressure ≥ 140/90 mm Hg), irrespective of treatment, and 1,541 (42.6%) of the participants were not prescribed any antihypertensive drug before participation in the HAI project. In this group, antihypertensive drug use after participation in HAI was related to blood pressure at baseline in HAI (Fig 4). Thus, in individuals with blood pressure of <130/80 mm Hg at baseline, 4.5% were after HAI prescribed at least 1 dose of antihypertensives, compared to 50.5% of individuals with a systolic blood pressure of at least 160 mm Hg or a diastolic blood pressure of at least 100 mm Hg at baseline (*p* < 0.001 for comparison). After HAI, we could track a total of 7,744 blood pressure measurements performed in 3,126 HAI participants, at general practitioners or in specialist healthcare. For individuals in HAI with a systolic blood pressure of less than 130 and a diastolic blood pressure of less than 80 mm Hg (*n* = 517) at baseline, the follow-up measurements showed that mean systolic blood pressure increased by 9.0 mm Hg (95% CI 7.4–10.6, *p* < 0.001), and mean diastolic blood pressure increased nonsignificantly by 0.4 mm Hg (95% CI −0.2 to 0.9, *p* = 0.16) during follow-up (Figs 5 and 6). In contrast, for participants with a systolic blood pressure of at least 160 mm Hg or a diastolic blood pressure of at least 100 mm Hg (*n* = 434) at baseline, mean systolic blood pressure decreased by 21.8 mm Hg (95% CI 19.8–23.8, *p* < 0.001), and mean diastolic blood pressure decreased by 9.6 mm Hg (95% CI 8.6–10.6, *p* < 0.001) after baseline (Figs 5 and 6).

In the HAI participants, 2,055 individuals (55.9%) were not treated with lipid-lowering therapy at baseline. In these individuals, lipid-lowering therapy after baseline was initiated based on LDL-cholesterol level at baseline (Fig 7). After baseline, 6,631 follow-up

**Table 3. Baseline characteristics of 70-year-old Umeå residents and 70-year-old residents from the rest of Sweden.**

| Variable | Baseline date 2006–2011 | | | Baseline date 2012–2017 | | |
|---|---|---|---|---|---|---|
| | Umeå residents | Sweden residents | SMD* | Umeå residents | Sweden residents | SMD* |
| *N* | 4,495 | 446,860 | | 6,665 | 650,699 | |
| **Age, mean (SD), years** | 70.5 (0.1) | 70.5 (0.1) | 0.02 | 70.5 (0.1) | 70.5 (0.1) | 0.00 |
| **Female sex, *n* (%)** | 2,369 (53.9) | 230,478 (51.6) | 0.05 | 3,371 (50.6) | 329,993 (50.7) | 0.00 |
| **Disposable income at age 60 years, mean (SD), 1,000 Swedish kronor** | 185 (230) | 179 (348) | 0.01 | 237 (366) | 235 (580) | 0.00 |
| Missing data, *n* | 0 | 148 | | 0 | 452 | |
| **Education[a], *n* (%)** | | | | | | |
| Primary | 1,326 (30.3) | 178,877 (40.0) | 0.23 | 1,400 (21.0) | 194,169 (29.8) | 0.22 |
| Secondary | 1,875 (42.8) | 168,665 (38.0) | 0.10 | 2,823 (42.4) | 276,745 (42.5) | 0.00 |
| Post-secondary | 1,179 (26.9) | 92,400 (20.7) | 0.15 | 2,429 (36.4) | 175,419 (27.0) | 0.20 |
| Missing data, *n* | 15 | 5,918 | | 13 | 4,366 | |
| **Civil status[a], *n* (%)** | | | | | | |
| Married | 2,840 (64.6) | 277,076 (62.0) | 0.06 | 4,047 (60.7) | 394,126 (60.6) | 0.00 |
| Never married | 323 (7.3) | 37,211 (8.3) | 0.04 | 794 (11.9) | 71,490 (11.0) | 0.03 |
| Widowed | 492 (11.2) | 49,763 (11.1) | 0.00 | 552 (8.3) | 55,376 (8.5) | 0.01 |
| Divorced | 740 (16.8) | 82,370 (18.4) | 0.05 | 1,272 (19.1) | 128,674 (19.8) | 0.02 |
| Other, *n* | 0 | 124 | | 0 | 266 | |
| Missing data, *n* | 0 | 316 | | 0 | 767 | |
| **Diagnoses, *n* (%)** | | | | | | |
| Stroke | 204 (4.6) | 17,614 (3.9) | 0.04 | 319 (4.8) | 27,018 (4.2) | 0.04 |
| Myocardial infarction | 163 (3.7) | 20,375 (4.6) | 0.05 | 352 (5.3) | 32,331 (5.0) | 0.02 |
| Heart failure | 116 (2.6) | 10,739 (2.4) | 0.02 | 174 (2.6) | 16,004 (2.5) | 0.01 |
| Angina pectoris | 383 (8.7) | 35,919 (8.0) | 0.03 | 540 (8.1) | 47,145 (7.2) | 0.04 |
| Diabetes | 451 (10.3) | 52,056 (11.6) | 0.05 | 820 (12.3) | 88,142 (13.5) | 0.05 |
| Fracture | 506 (11.5) | 45,364 (10.2) | 0.05 | 1,132 (17.0) | 95,765 (14.7) | 0.08 |
| Rheumatoid arthritis | 95 (2.2) | 7,638 (1.7) | 0.04 | 142 (2.1) | 12,556 (1.9) | 0.02 |
| Chronic obstructive pulmonary disease | 131 (3.0) | 10,808 (2.4) | 0.04 | 185 (2.8) | 18,788 (2.9) | 0.01 |
| Renal failure | 39 (0.9) | 4,007 (0.9) | 0.00 | 91 (1.4) | 9,347 (1.4) | 0.01 |
| Crohn disease | 19 (0.4) | 2,327 (0.5) | 0.01 | 48 (0.7) | 5,181 (0.8) | 0.01 |
| Ulcerative colitis | 28 (0.6) | 2,645 (0.6) | 0.01 | 48 (0.7) | 5,786 (0.9) | 0.03 |
| Parkinson disease | 36 (0.8) | 2,613 (0.6) | 0.03 | 56 (0.8) | 4,097 (0.6) | 0.03 |
| Dementia | 80 (1.8) | 4,726 (1.1) | 0.06 | 85 (1.3) | 6,794 (1.0) | 0.03 |
| Depression | 695 (15.8) | 66,112 (14.8) | 0.03 | 1,525 (22.9) | 143,218 (22.0) | 0.03 |
| Bipolar disorder | 11 (0.3) | 1,968 (0.4) | 0.04 | 48 (0.7) | 4,072 (0.6) | 0.02 |
| Alcohol intoxication | 63 (1.4) | 6,873 (1.5) | 0.01 | 129 (1.9) | 15,237 (2.3) | 0.04 |
| Opioid intoxication | 1 (0.02) | 203 (0.05) | 0.02 | 3 (0.05) | 650 (0.1) | 0.03 |
| Cancer | 652 (14.8) | 66,400 (14.9) | 0.00 | 1,257 (18.9) | 124,917 (19.2) | 0.01 |
| **Medications[b], *n* (%)** | | | | | | |
| Antihypertensive | 2,455 (55.9) | 210,518 (47.1) | 0.19 | 4,072 (61.1) | 355,369 (54.6) | 0.14 |
| Lipid-lowering agent | 1,589 (36.2) | 144,738 (32.4) | 0.09 | 2,990 (44.9) | 254,797 (39.2) | 0.11 |
| Anticoagulant | 1,607 (36.6) | 153,512 (34.4) | 0.05 | 2,803 (42.1) | 255,797 (39.3) | 0.06 |
| Neuroleptic | 162 (3.7) | 12,565 (2.8) | 0.05 | 271 (4.1) | 20,604 (3.2) | 0.05 |
| Hypnotic | 1,004 (22.8) | 94,302 (21.1) | 0.05 | 1,861 (27.9) | 175,890 (27.0) | 0.02 |
| Sedative | 360 (8.2) | 63,296 (14.2) | 0.23 | 848 (12.7) | 133,387 (20.5) | 0.23 |

(*Continued*)

**Table 3.** (Continued)

| Variable | Baseline date 2006–2011 | | | Baseline date 2012–2017 | | |
|---|---|---|---|---|---|---|
| | Umeå residents | Sweden residents | SMD* | Umeå residents | Sweden residents | SMD* |
| Immunosuppressant | 122 (2.8) | 9,309 (2.1) | 0.05 | 247 (3.7) | 20,220 (3.1) | 0.03 |

Data are presented for those with a baseline date before (years 2006–2011) and after (years 2012–2017) the Healthy Ageing Initiative program started.

*Standardized mean difference.

[a]Education and civil status recorded in the calendar year before the baseline date.

[b]Prescriptions filled since July 2005.

measurements of LDL-cholesterol were obtained in 2,347 individuals from the HAI cohort. As was the case for blood pressure, LDL-cholesterol levels were reduced depending on the LDL-levels at baseline (Fig 8). For those with LDL-cholesterol levels above 4.11 mmol/l at baseline, a mean reduction of 1.3 mmol/l was seen more than 2 years after the initial measurement. In contrast, LDL-cholesterol did not change in those with LDL-cholesterol < 3.36 mmol/l at baseline.

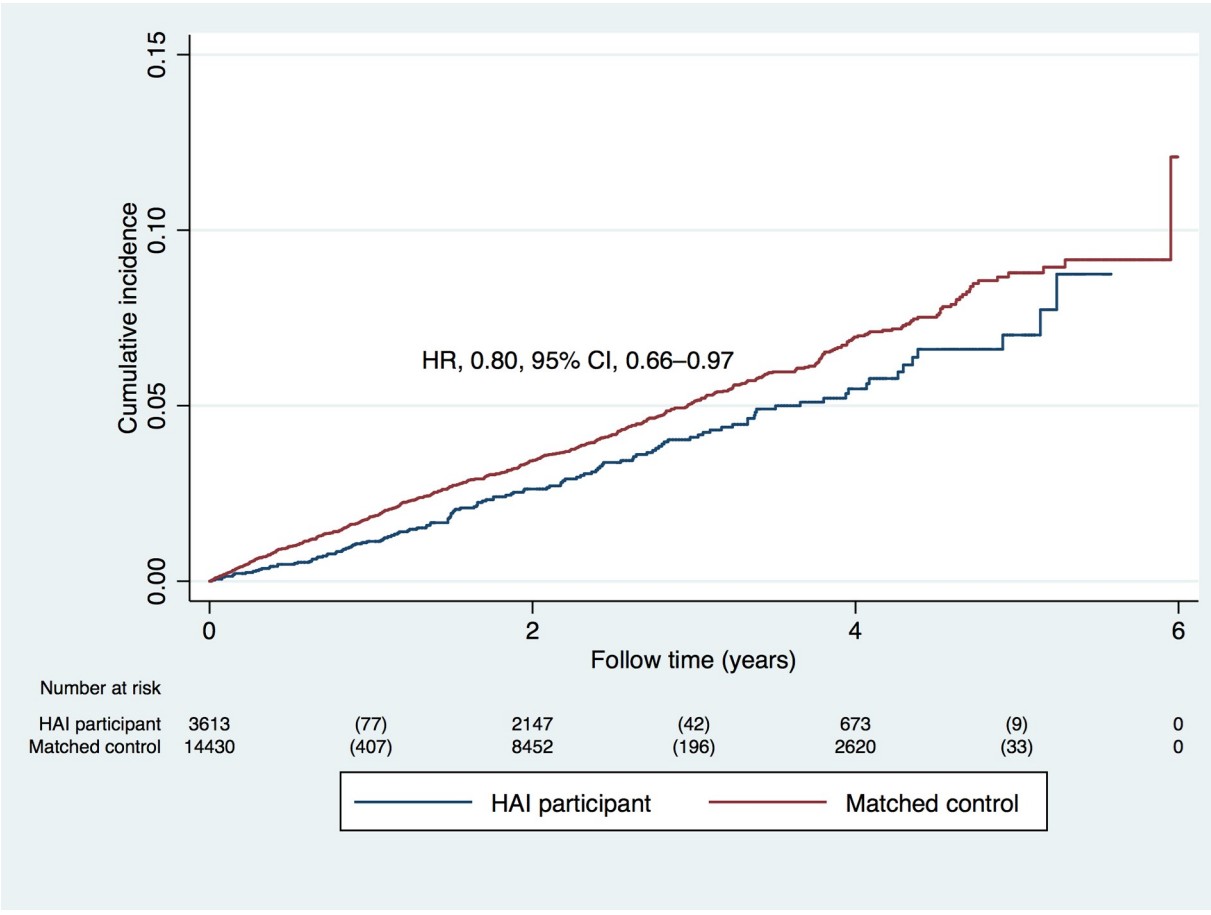

**Fig 1. Risk of stroke, myocardial infarction, or angina pectoris in participants of the Healthy Ageing Initiative (HAI, *n* = 3,613) and matched controls (*n* = 14,452).** The hazard ratio (HR) is presented for the time to first outcome, and below the figure the number at risk at each time point is presented, together with the number of outcome events within parentheses.

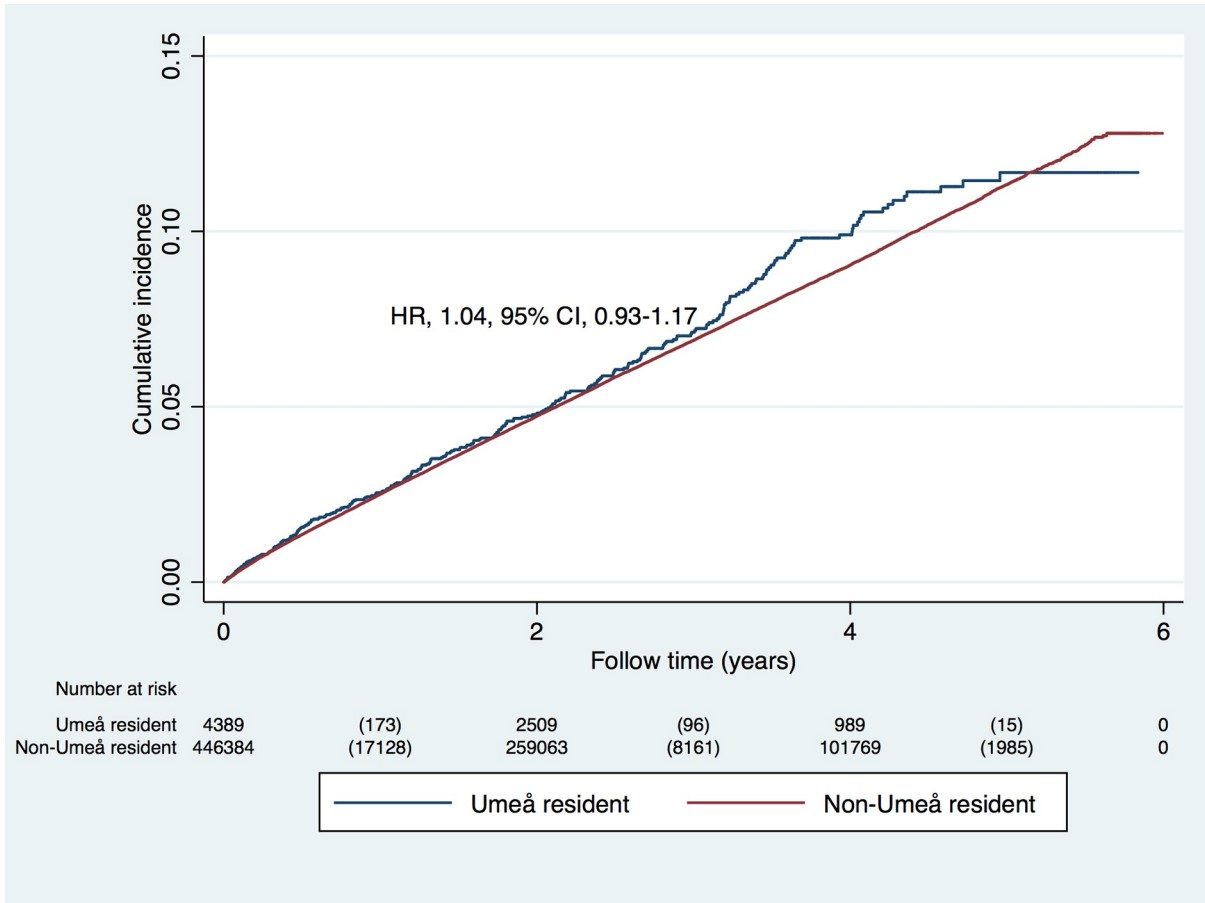

**Fig 2. Risk of stroke, myocardial infarction, or angina pectoris in 70-year-old Umeå residents and in 70-year-olds from the rest of Sweden with baseline date in 2006–2011.** The hazard ratio (HR) is presented for time to first outcome, and below the figure the number at risk at each time point is presented, together with the number of outcome events within parentheses.

## Discussion

In community-dwelling 70-year-olds, a multidimensional prevention program was associated with a 20% lower risk of CVD during follow-up. Consistent with the main results, the risk of CVD was lower in 70-year-olds in the prevention program municipality than in the rest of Sweden after, but not before, the HAI project was initiated. An analysis of intermediate outcomes in HAI participants showed that detected hypertension and high blood lipids at baseline were associated with initiation of therapy, and greater reductions in these risk factors during follow-up.

To our knowledge, no large randomized or observational study has evaluated the effects of multidimensional prevention programs on CVD in people aged 65 years or older. In people aged 50 years on average, a meta-analysis of randomized intervention studies found no effect of such programs on all-cause mortality or coronary heart disease mortality [4]. The associations found in the present project could be related to the participants' older age and greater number of risk factors at baseline. In support of this hypothesis, the mentioned meta-analysis did find significant effects on all-cause mortality and CVD in participants with diabetes or hypertension [4]. It has also been demonstrated previously that the importance of risk factors for CVD increase with increasing age [8].

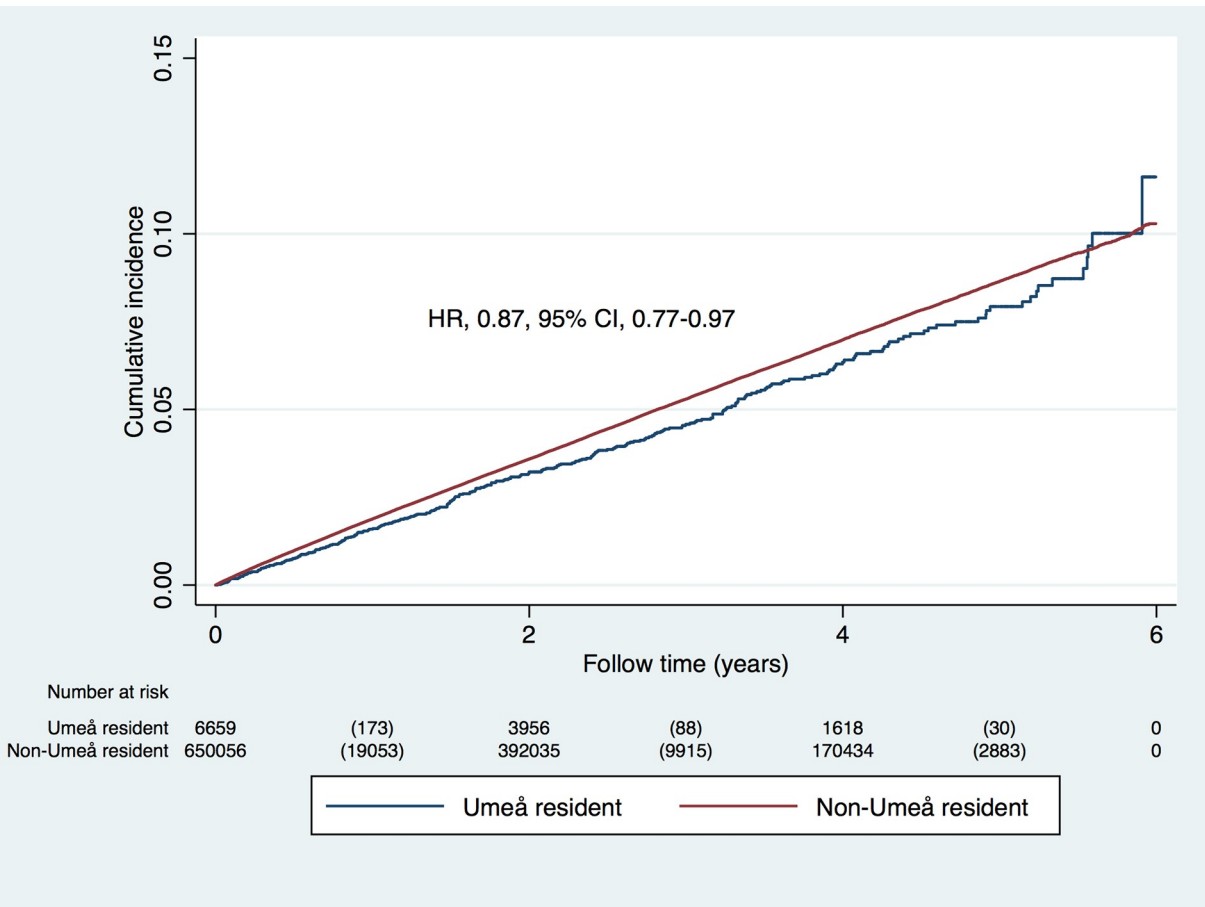

**Fig 3. Risk of stroke, myocardial infarction, or angina pectoris in 70-year-old Umeå residents and in 70-year-olds from the rest of Sweden with baseline date in 2012–2017.** The hazard ratio (HR) is presented for time to first outcome, and below the figure the number at risk at each time point is presented, together with the number of outcome events within parentheses.

In addition to the risk factor burden in the study population, the success of a primary prevention program may be determined by the risk factor the program is targeting, and if the program is modifying a single risk factor or the total risk factor burden. Recently, the Look AHEAD Research Group found that an intervention program targeting weight loss had no effect on death or CVD in a large randomized study of overweight and obese individuals with type 2 diabetes [9]. According to the authors, one reason for the lack of effect could be good medical management of risk factors for CVD in primary care, i.e., risk factors other than obesity. This is an important point, since the most important risk factor for CVD and mortality is probably hypertension [10–12]. In the Look AHEAD study population, systolic blood pressure was on average 130 mm Hg at baseline [9], suggesting a rather good adherence to current knowledge and guidelines [13], which likely influenced the chance to show an effect of the intervention. In our population, about 50% had stage 2 hypertension at baseline [14], irrespective of pharmacological treatment, suggesting excellent opportunities for improved blood pressure control.

As such, we were interested in also investigating changes in blood pressure and the use of blood pressure medications after the HAI prevention program. In participants with baseline blood pressure below 130/80 mm Hg, 4.5% were dispensed a hypertension drug for the first time during follow-up. In contrast, about 50% of participants were prescribed an antihypertensive for the first time after baseline if they had a systolic blood pressure of at least 160 mm Hg

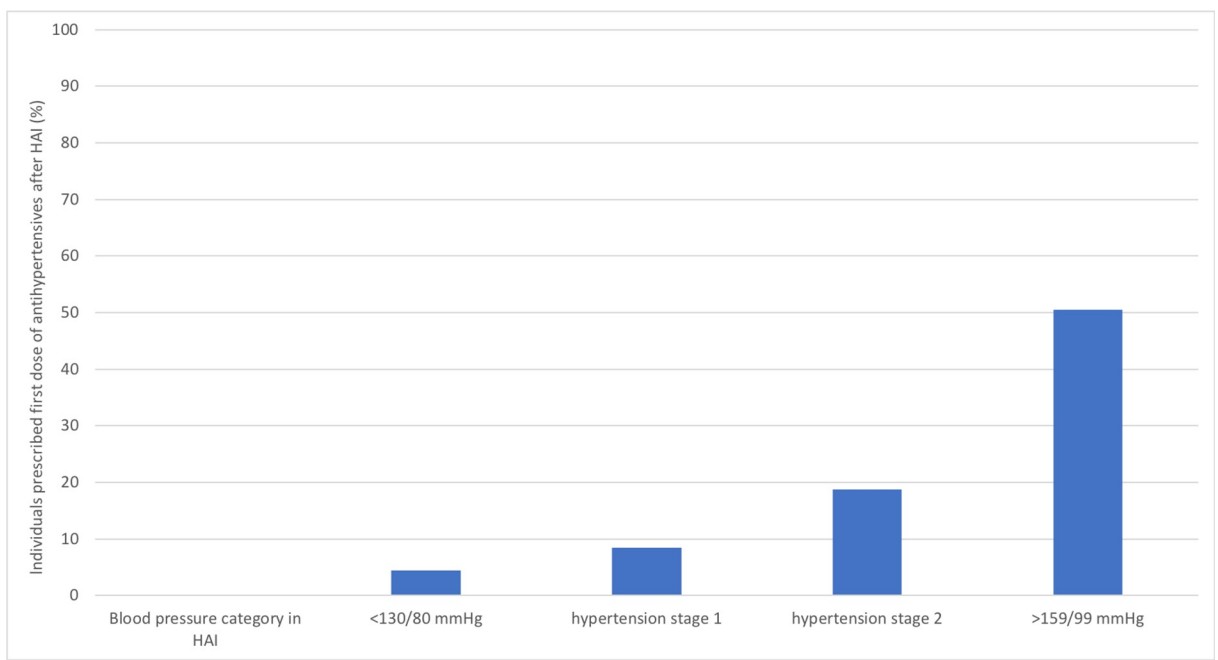

**Fig 4. Initiation of blood pressure treatment in treatment-naïve Healthy Ageing Initiative (HAI) participants based on blood pressure category at baseline (*n* = 1,541).** Hypertension stage 1: systolic blood pressure of 130–139 mm Hg or diastolic blood pressure of 80–89 mm Hg. Hypertension stage 2: systolic blood pressure of ≥140 mm Hg or diastolic blood pressure of ≥90 mm Hg.

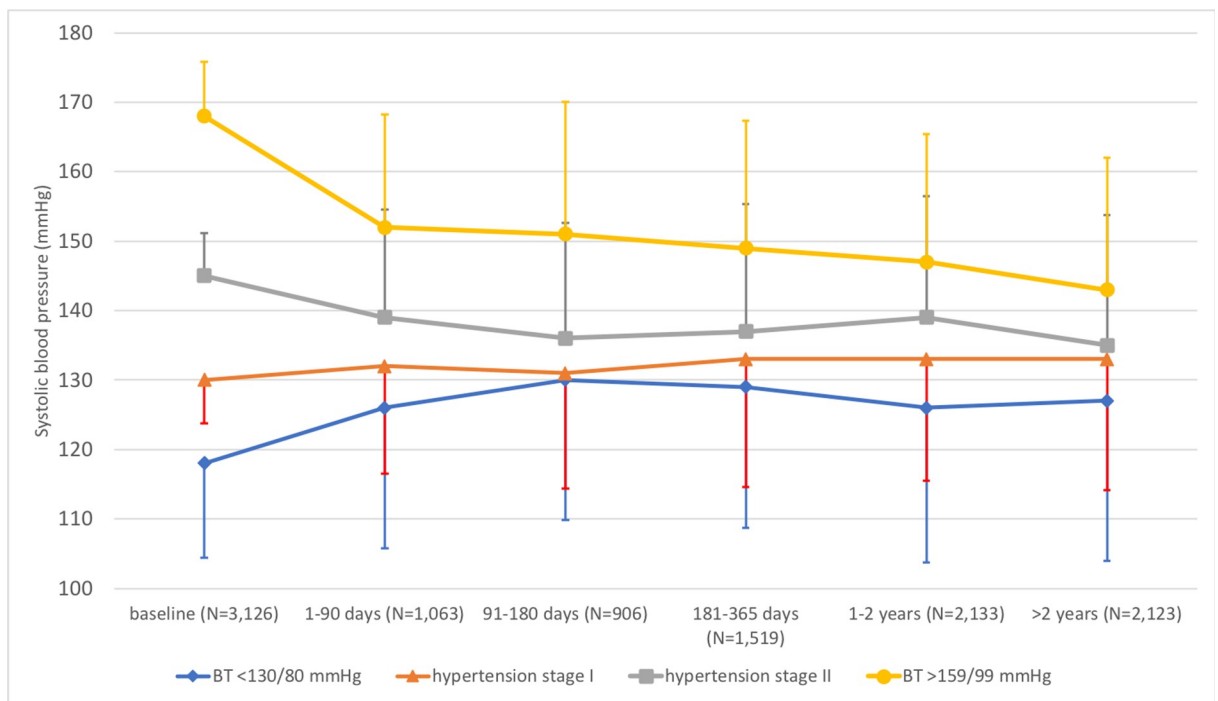

**Fig 5. Changes in systolic blood pressure after the baseline examination in Healthy Ageing Initiative (HAI) participants, based on blood pressure category at baseline.** The figure is based on a total of 7,744 follow-up measurements in 3,126 HAI participants. Means and standard deviations are presented. Hypertension stage 1: systolic blood pressure of 130–139 mm Hg or diastolic blood pressure of 80–89 mm Hg. Hypertension stage 2: systolic blood pressure ≥140 mm Hg or diastolic blood pressure of ≥90 mm Hg. BT, blood pressure.

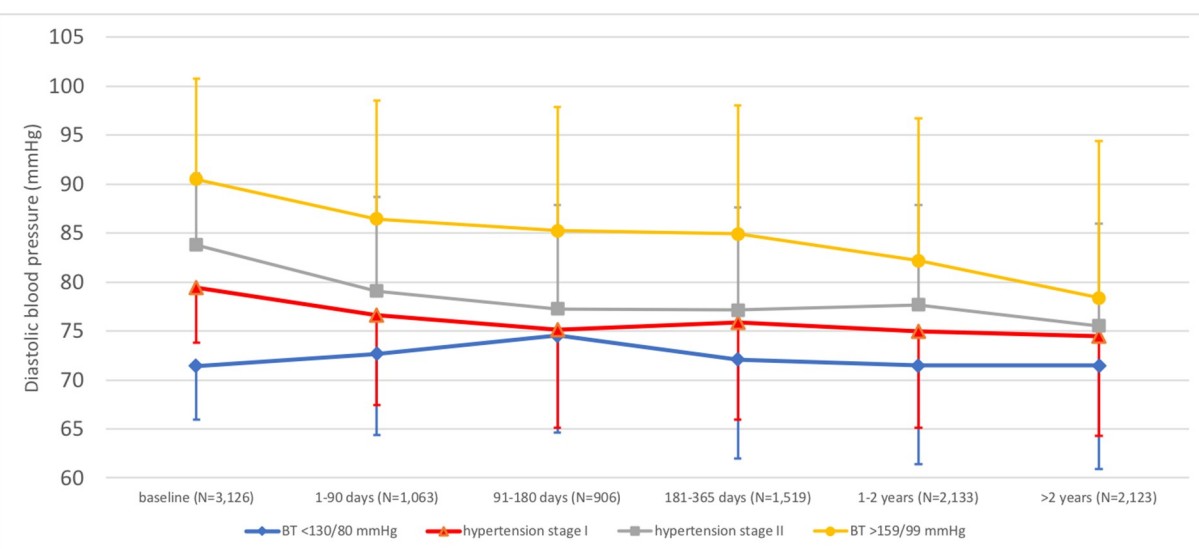

**Fig 6. Changes in diastolic blood pressure after the baseline examination in Healthy Ageing Initiative (HAI) participants, based on blood pressure category at baseline.** The figure is based on a total of 7,744 follow-up measurements in 3,126 HAI participants. Means and standard deviations are presented. Hypertension stage 1: systolic blood pressure of 130–139 mm Hg or diastolic blood pressure of 80–89 mm Hg. Hypertension stage 2: systolic blood pressure of ≥140 mm Hg or diastolic blood pressure of ≥90 mm Hg. BT, blood pressure.

or a diastolic blood pressure of at least 100 mm Hg. These prescription patterns, favoring those with higher blood pressure, were accompanied by a blood pressure reduction during follow-up, especially for those with severe hypertension at baseline. Thus, in participants with a blood

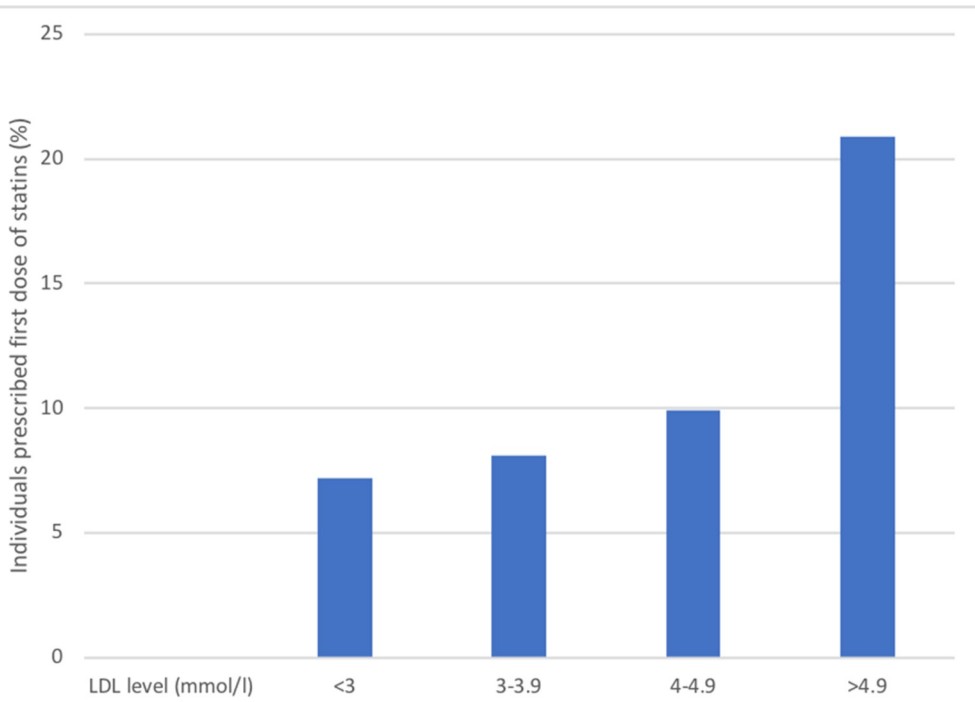

**Fig 7. Initiation of lipid-lowering therapy (statin) in treatment-naïve Healthy Ageing Initiative (HAI) participants based on low-density lipoprotein (LDL)–cholesterol levels obtained at baseline (*n* = 2,055).**

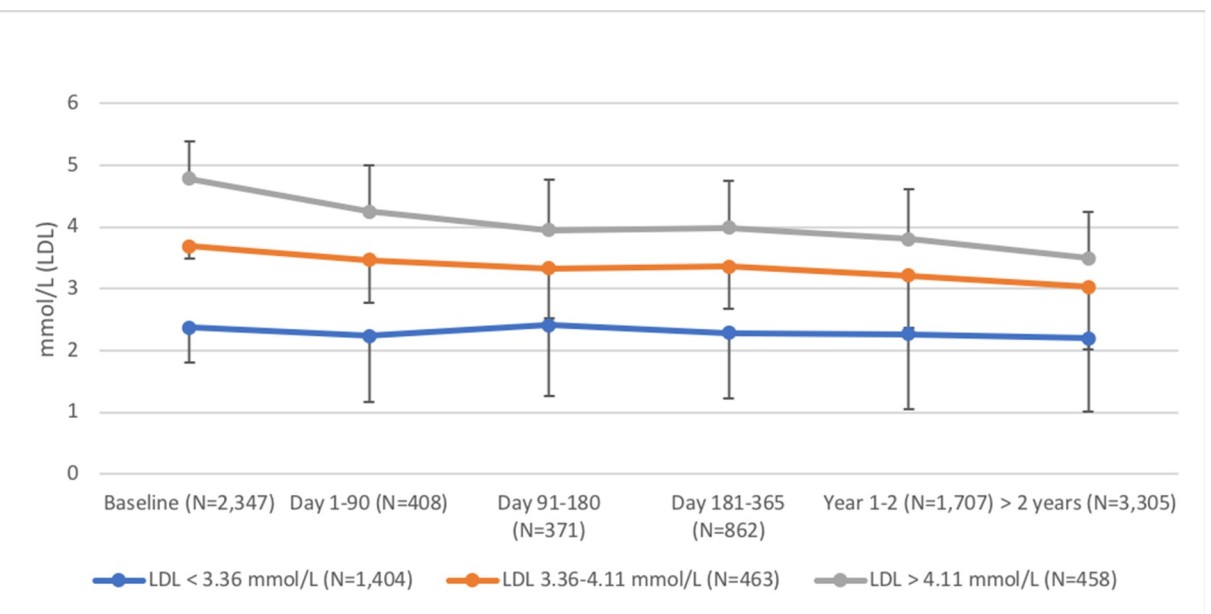

**Fig 8. Changes in low-density lipoprotein (LDL)–cholesterol levels after the baseline examination in Healthy Ageing Initiative (HAI) participants, based on LDL-cholesterol at baseline.** The figure is based on a total of 6,631 measurements in 2,347 HAI participants. Participants were categorized in 3 groups based on LDL-cholesterol level at baseline: optimal (<3.36 mmol/l, blue line), intermediate (3.36–4.11 mmol/l, orange line), and high (>4.11 mmol/l, gray line). Means and standard deviations are presented.

pressure of at least 160/100 mm Hg, there was a fast reduction of blood pressure after baseline, and based on all measurements, a mean reduction in systolic blood pressure of 22 mm Hg during follow-up. We also evaluated whether the prevention program was associated with improved control of hypercholesterolemia. Similar to the findings for blood pressure, treatment with lipid-lowering agents was initiated after baseline based on the lipid levels at the initial investigation within HAI. Furthermore, LDL-cholesterol during follow-up decreased based on these prescription patterns. In individuals with LDL-cholesterol levels above 4.11 mmol/l at baseline, a mean reduction of 1.3 mmol/l was seen more than 2 years after the initial measurement. In contrast, for those with optimal lipid levels at baseline, no changes were seen during follow-up. Based on the results of randomized controlled studies, it is quite clear that such reductions in blood pressure and blood lipids are associated with a substantial risk reduction for CVD [15].

Studies have indicated that risk factors for CVD, such as blood pressure and diabetes, are better at predicting disease in older than in younger age groups [16–18], probably in part because older people have more risk factors. Since interventions are usually more effective in groups with many risk factors [19], these are good reasons to include older people in primary prevention programs. Of our participants, 6% were smokers, the majority were overweight, about 50% had stage 2 hypertension at baseline irrespective of treatment, and 25% had diabetes or fasting glucose impairment. Yet, this burden of risk factors is modest from a global perspective. According to the American Heart Association, more than 60% of Americans aged 65–74 years have hypertension [20]. In Europe, a recent study of 17 countries showed that 11.5% of Europeans are smokers [21]. Thus, any effects of our prevention program were likely not related to an unusually high risk factor burden in the study population. The above-mentioned studies also indicate that it would be of high interest to investigate the effects of preventive measures on the risk of CVD in older people from other countries with an even higher risk factor burden.

Although the risk factor burden in our population may be regarded as modest in an international perspective, it would have been unethical to randomize participants to the prevention program. The disadvantage of not randomizing is that the associations found may be explained by confounding, although participants and controls were matched closely on many potential confounders at baseline. To investigate further the possibility of confounding, we performed a sensitivity analysis that showed that the risk of stroke or ischemic heart disease was similar in Umeå Municipality and the rest of Sweden prior to the start of the prevention program. Thus, 70-year-old Umeå residents overall were probably not healthier than controls due to self-selection, as the presence of CVD was similar in Umeå and the rest of Sweden before the HAI project started. In contrast, the risk of ischemic heart disease and stroke was 13% lower in 70-year-olds living in Umeå after the start of the prevention program—in which 54% of 70-year-olds living in Umeå Municipality participated. The consistent results of these analyses suggest that confounding does not explain our findings. Different components of the prevention program may have contributed to the lower risk of CVD found in HAI participants: There may have been improved medication using drugs known to reduce the risk of CVD, and the motivational interview, including advice with respect to food intake and increased physical activity, may have contributed to the lower risk of CVD. Interestingly, a recent meta-epidemiological study suggested that exercise and drug interventions have similar mortality benefits and effects in the secondary prevention of ischemic heart disease [22]. Future studies are needed to investigate whether these findings are generalizable to other populations of community-dwelling older people.

There are several limitations of the present study that should be acknowledged. In particular, this is an observational study, and the associations found are not proof of causal effects. However, as explained above, given the risk factor burden, a randomized trial in this population could not have been performed due to ethical reasons. We investigated changes in the intermediate endpoints blood pressure and lipid levels to evaluate whether the lower risk of CVD in the HAI participants could be explained by changes in these risk factors. As discussed above, reductions in both blood pressure and lipid levels were seen, especially in those with higher values at baseline. Although this may support the main association found with respect to reduced risk of CVD, these changes were also most likely influenced by regression towards the mean. Therefore, we can only speculate as to whether any effects on CVD are related to changes in blood pressure and lipid levels from improved medication and/or from behavior changes from the motivational interview. The strengths of the study include the large sample of 70-year-olds included in this study and endpoints of high relevance, captured with high precision in national registers with a low loss to follow-up. Given that the cohort investigated is population based, the results are likely generalizable to other 70-year-old men and women.

In summary, a multidimensional primary prevention program was associated with a reduced risk of ischemic heart disease and stroke in community-dwelling 70-year-olds. The prevention program was also associated with improved treatment of hypertension and hypercholesterolemia, particularly in participants at higher risk. Given that the risk factor burden was modest compared to in other countries, it would be interesting and important to see evaluations of similar programs elsewhere. Since the world's population is ageing, primary prevention will probably play a key role in healthcare in the future.

## Supporting information

**S1 STROBE Checklist.**
(DOC)

**S1 Appendix. Description of the prevention program given within the Healthy Ageing Initiative.**
(DOCX)

**S1 Table. Variable definitions.**
(DOCX)

## Acknowledgments

The authors would like to thank the research staff participating in the project, Magnus Lindblom, David Lapveteläinen, Jim Viklund, and Monica Rasmussen Ahlenius.

## Author Contributions

**Conceptualization:** Anna Nordström, Bo Carlberg, Peter Nordström.

**Formal analysis:** Anna Nordström, Jonathan Bergman, Andreas Hult, Peter Nordström.

**Funding acquisition:** Anna Nordström, Peter Nordström.

**Investigation:** Sabine Björk, Bo Carlberg, Jonas Johansson, Andreas Hult.

**Methodology:** Anna Nordström, Sabine Björk, Jonas Johansson, Peter Nordström.

**Project administration:** Anna Nordström, Peter Nordström.

**Resources:** Andreas Hult.

**Supervision:** Anna Nordström, Bo Carlberg, Peter Nordström.

**Validation:** Anna Nordström, Jonas Johansson.

**Writing – original draft:** Anna Nordström, Jonathan Bergman, Sabine Björk, Peter Nordström.

**Writing – review & editing:** Anna Nordström, Jonathan Bergman, Sabine Björk, Bo Carlberg, Jonas Johansson, Andreas Hult, Peter Nordström.

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
