## [Decision Letter · Decision Letter 0]

11 Mar 2020

Dear Dr. Nordström,

Thank you very much for submitting your manuscript 'A multiple risk factor intervention is associated with decreased risk of cardiovascular disease in 70-year-olds: A Cohort Study' (PMEDICINE-D-19-03481) for consideration at PLOS Medicine. I sincerely apologise for the delay to your submission as we had some difficulty securing reviewers. I hope you will find their comments useful. 

Your paper was evaluated by a senior editor and discussed among all the editors here. It was also sent to independent peer reviewers, whose comments you can read at the bottom of this email. Any accompanying reviewer attachments can be seen via the link below:

[LINK]

We would like to consider a revised version that addresses the reviewers' and editors' comments. Please note that we cannot make any decision about publication until we have seen the revised manuscript and your response, and we may seek re-review by one or more of the reviewers. 

In revising the manuscript for further consideration here, your revisions should address the specific points made by each reviewer and the editors. Please also check the guidelines for revised papers at http://journals.plos.org/plosmedicine/s/revising-your-manuscript for any that apply to your paper. In your rebuttal letter you should indicate your response to any comments from reviewers or editors and the changes you have made in the manuscript. Please submit a clean version of the paper as the main article file; a version with changes marked should be uploaded as a marked up manuscript. 

In addition, we request that you upload any figures associated with your paper as individual TIF or EPS files with 300dpi resolution at resubmission; please read our figure guidelines for more information on our requirements: http://journals.plos.org/plosmedicine/s/figures.

While revising your submission, please upload your figure files to the PACE digital diagnostic tool, https://pace.apexcovantage.com/ PACE helps ensure that figures meet PLOS requirements. To use PACE, you must first register as a user. Then, login and navigate to the UPLOAD tab, where you will find detailed instructions on how to use the tool. If you encounter any issues or have any questions when using PACE, please email us at PLOSMedicine@plos.org.

We expect to receive your revised manuscript by Mar 31 2020 11:59PM. Please email us (plosmedicine@plos.org) if you have any questions or concerns.

Your article can be found in the 'Submissions Needing Revision' folder. 

We look forward to receiving your revised manuscript. 

Sincerely,

Adya Misra, PhD

Senior Editor 

PLOS Medicine

plosmedicine.org

Title- please consider amending your title to avoid the use of “intervention” as this suggests the study is a clinical trial or related to one. 

Abstract- please consider providing additional background information to place the study into context 

Abstract methods and findings-please provide brief participant demographics

Abstract methods and findings- the last sentence should highlight the limitation of your study design

On page 3- please include the heading “Author Summary”

Author summary- please revise to clarify that the authors did not undertake the health survey themselves as it currently reads as an intervention from a clinical trial. Please also provide exact p-values 

References- please place square brackets after the full stop for example: [3]. 

Introduction- Line 5 paragraph 1, we suggest “fortunately” is removed here

Introduction-Line 5 paragraph 2 lack of “effect” instead of “effects”

Page 4 last sentence- please rephrase this to sound less like a clinical trial and more as a cohort study

Methods- if there is a reason to choose participants exactly at the age of 70, could you please mention this 

Methods- please report the study in line with STROBE guidelines and provide the completed checklist as supplementary information. In the methods, please mention the study has been reported according to the STROBE guideline and the checklist can be found as SI file xx. The STROBE guideline can be found here: http://www.equator-network.org/reporting-guidelines/strobe/ When completing the checklist, please use section and paragraph numbers, rather than page numbers.

For all observational studies, in the manuscript text, please indicate: (1) the specific hypotheses you intended to test, (2) the analytical methods by which you planned to test them, (3) the analyses you actually performed, and (4) when reported analyses differ from those that were planned, transparent explanations for differences that affect the reliability of the study's results. If a reported analysis was performed based on an interesting but unanticipated pattern in the data, please be clear that the analysis was data-driven.

Did your study have a prospective protocol or analysis plan? Please state this (either way) early in the Methods section.

Please provide the questionnaires used as part of the intervention and also describe the motivational interview in greater detail. Please describe who undertook the interviews, which language were this conducted in, were there any prepared questions or were these open ended. 

Please provide brief details of HLA, including follow-up visits, medication prescribing etc

Please provide details of informed consent in the methods section where you note the ethics approval received

Please provide p-values along with all 95% confidence intervals and vice versa. Please provide exact p-values unless p<0.001 

Please note we do not permit instances of data not shown as per PLOS data policy (on page 10). Please provide these data or remove this reference. 

Page 12 second paragraph- please revise “hypertension drugs”

We understand that the primary data cannot be shared due to legal and ethical restrictions. Authors do not need to submit their entire data set, or the raw data collected during an investigation. Please submit the following data: 

The values behind the means, standard deviations and other measures reported; 

The values used to build graphs; 

Comments from the reviewers:

Reviewer #1: I confine my remarks to statistical aspects of this paper. These were very well done and I have only some minor comments.

NOTE: Line numbers would have made the review easier

p. 2 First para. Not staitistical but ... prevention of what?

p. 4 Another reason that other studies may have shown small effects is limited follow up time. i.e. Changes that a 40 year old makes may affect his or her chances of having a heart attack at 60 or 70, but few studies are on that time frame.

p. 7 Why was a quadratic for income added? Why not a spline?

p. 9 Bottom - Which number had to be higher? Systolic, diastolic, both?

Table 2 - you need to say what the numbers are (mean and sd? Or what)

Figure 4 (several) if you add jitter, it will be easier to see what is going on.

Reviewer #2: This manuscript evaluated the prevention effects of a multidimensional CVD prevention program, the Healthy Ageing Initiative (HAI), among a community-dwelling senior population who were 70 years old at baseline. The intervention effectiveness was evaluated using two approaches, a propensity score matched case control design and an intention to treat analysis comparing all Umea residents with 70-year-olds who live in other parts of Sweden. Both designs showed the intervention significantly reduced the CVD risk among those who participated in the intervention. 

In general, this study is well designed and clearly written. However, I have a few concerns that need to be addressed.

Major Concerns:

1. My major concern for this study is that the analysis to examine the intervention effects on the control of blood pressure and lipid levels was not done properly. Specifically: 

1) First, the intervention effects on these outcomes were only assessed among the HAI participants, but not in the controls. However, without comparing the outcomes between cases and controls, it is hard to conclude that the improvements in blood pressure and lipid levels were caused by the intervention. 

2) The authors showed that systolic blood pressure among HAI participants increased over time in those with normal baseline blood pressure but decreased among those with very high blood pressure at baseline. This observation is likely caused by "regression toward the mean", at least partially. 

3) The changes in blood pressure and lipid levels were assessed using the mean and standard deviations among those who had data at each time point. As the sample sizes changed a lot at different time points, the raw means at different time points are not directly comparable. To make the conclusions more convincible, it is important to compare the means at different time points using a longitudinal model, such as linear mixed model. 

2. Another major concern I have is that the authors did not include a limitation subsection in the Discussion section. Although several paragraphs in the Discussion section described some limitations of the current study, it will be helpful to put all the potential limitations together for the readers to realize these issues fairly easily. 

Minor Concerns:

1. Page 9, last paragraph: It is unclear what percentages they report in the last paragraph on this page. For example, what's the denominator for the percentages shown in the second line of this paragraph?

2. Page 13, first line: What are the other risk factors mentioned in the first sentence? 

3. Page 14, 1st paragraph, line 5-6: The meaning of the sentence started with "Different components" is unclear and needs to be clarified. 

4. Page 14, 1st paragraph, line 6: It's better to change "possibility" to "opportunity" in this sentence. 

5. Figure 2a and Figure 2b can be merged into one figure. 

Reviewer #3: In this manuscript entitled "A multiple risk factor intervention is associated with decreased risk of cardiovascular disease in 70-year-olds: A Cohort Study", the authors present findings from a cohort study in which the effect of a program aimed at cardiovascular primary prevention on a composite outcome of stroke, myocardial infarction and angina. The study included adults aged 70 living in Sweden. The intervention included a review of the participant's cardiovascular risks along with motivational interviewing techniques on cardiovascular risk reduction. The authors used propensity score matching and during a mean follow up of 2.5 years, identified a reduced risk of the primary outcome in participants who received the intervention. 

This study and analysis was well designed and the authors recruited a large number of participants in a single site. I have a few points for the authors to consider.

Major

Abstract

1. Please state the aim of the study more clearly. Consider including the primary outcome and more detail on the nature of the primary prevention program.

Introduction

1. Consider stating the hypothesis and aim of this study explicitly in the introduction

2. Consider explaining the "multiple risk factor intervention" in more detail in this section

Methods

1. Was a sample size estimation completed for this study?

2. Did the authors consider conducting a randomized controlled trial and what were the reasons for choosing a cohort study design?

3. 54% of residents enrolled in the study. Is it known why 46% of residents did not participate and are demographic details available for this cohort?

Results

1. Is it possible the observed result was due solely to prescription of lipid lowering therapy and antihypertensives and not the intervention?

2. There were higher rates of antihypertensive and lipid lowering therapies in the Umea general population. Was there a difference in prescribing rates between Umea residents who were and were not participants in the study?

Discussion

1. This study found that participants were more likely to be prescribed blood pressure and lipid lowering therapy by their primary care doctor - can the authors elaborate further on how the motivational interview techniques conferred benefit in addition to this?

Minor

Methods

1. Consider stating the study type early in the Methods section eg. cohort study

2. Stroke was included in the composite primary outcome. Was this ischemic, hemorrhagic stroke or both?

3. "Feedback is given continuously" - can you describe in more detail what form this feedback took?

References

1. Citation number 8 is invalid

[LINK]

---

## [Decision Letter · Decision Letter 1]

15 Apr 2020

Dear Dr. Nordström,

Thank you very much for re-submitting your manuscript "A multiple risk factor program is associated with decreased risk of cardiovascular disease in 70-year-olds: A Cohort Study" (PMEDICINE-D-19-03481R1) for review by PLOS Medicine.

I have discussed the paper with my colleagues and the academic editor and it was also seen again by 3 reviewers. I am pleased to say that provided the remaining editorial and production issues are dealt with we are planning to accept the paper for publication in the journal.

[LINK]

We look forward to receiving the revised manuscript by Apr 22 2020 11:59PM. 

Sincerely,

Adya Misra, PhD

Senior Editor 

PLOS Medicine

plosmedicine.org

Comments from the Academic Editor:

I would like to see the authors make a more explicit statement in the abstract and in the limitations sections that the BP and LDL improvements are very likely influenced by regression to the mean as pointed out by one of the reviewers. I think they overstate the potential benefits of these reductions when we have no way of knowing what happened in the control group.

On the prescription data I may be missing something but why is this information not available for the control group from the Prescribed Drug Register. If there was an overall increase in prescription rate and/or drug dispensing rates over the follow-up period in the HAI group, that would be stronger evidence to support the assertion that the program did drive better BP and LDL control in the absence of these data in the control group. I appreciate that this may not be able to be disaggregated by baseline lipid and BP levels as done in Fig 3 and 5 for the control group but an overall comparison would add considerable information. I also think that looking at dispensing of scripts as it is a reasonably robust measure of adherence.

xxx

Requests from Editors:

Title- Could we add Sweden to the title?

Abstract

Please include limitations of your study design as the last sentence of the methods and findings section

Abstract Conclusions:

* Please address the study implications without overreaching what can be concluded from the data; the phrase "In this study, we observed ..." may be useful.

* Please interpret the study based on the results presented in the abstract, emphasizing what is new without overstating your conclusions.

* Please avoid vague statements such as "these results have major implications for policy/clinical care". Mention only specific implications substantiated by the results.

* Please avoid assertions of primacy ("We report for the first time....")

Lines 13-16 – please replace “intervention” with a more appropriate word 

Please briefly include limitations in the author summary section “what do these findings mean”

Methods

Line 5, 7 please rephrase “intervention program”. 

STROBE checklist- please remove page numbers as these are likely to change during publication

Discussion 

Several instances of “intervention”. Please rephrase

Comments from Reviewers:

Reviewer #1: The authors have addressed my concerns and I now recommend publication.

My only issue is their response to my last question -- I don't know how this works at PLOS and so, I leave it to the editors.

Peter Flom

Reviewer #2: This revision has substantially improved the manuscript and addressed all of my previous concerns.

Reviewer #3: In this revised manuscript entitled "A multiple risk factor program is associated with decreased risk of cardiovascular disease in 70-year-olds: A Cohort Study", the authors addressed the points raised by the reviewers.

This study and analysis was well designed. However, the observed outcomes have several limitations and I do not feel the findings of this study are novel or impact on direct patient care enough to recommend publication.

[LINK]

---

## [Editor Report · Decision Letter 2]

11 May 2020

Dear Dr. Nordström, 

On behalf of my colleagues and the academic editor, Dr. David Peiris, I am delighted to inform you that your manuscript entitled "A multiple risk factor program is associated with decreased risk of cardiovascular disease in 70-year-olds: A cohort study from Sweden" (PMEDICINE-D-19-03481R2) has been accepted for publication in PLOS Medicine. 

PRODUCTION PROCESS

PRESS

PROFILE INFORMATION

Thank you again for submitting the manuscript to PLOS Medicine. We look forward to publishing it. 

Best wishes, 

Adya Misra, PhD

Senior Editor 

PLOS Medicine

plosmedicine.org